# Electrical Discharge Machining of Alumina Using Cu-Ag and Cu Mono- and Multi-Layer Coatings and ZnO Powder-Mixed Water Medium

Anna A. Okunkova [1,]* , Marina A. Volosova [1], Khaled Hamdy [1,2] and Khasan I. Gkhashim [1]

1    Department of High-Efficiency Processing Technologies, Moscow State University of Technology STANKIN, Vadkovsky per. 1, 127994 Moscow, Russia
2    Production Engineering and Mechanical Design Department, Faculty of Engineering, Minia University, Minia 61519, Egypt
*    Correspondence: a.okunkova@stankin.ru; Tel.: +7-909-913-12-07

**Abstract:** The paper aims to extend the current knowledge on electrical discharge machining of insulating materials, such as cutting ceramics used to produce cutting inserts to machine nickel-based alloys in the aviation and aerospace industries. Aluminum-based ceramics such as $Al_2O_3$, AlN, and SiAlON are in the most demand in the industry but present a scientific and technical problem in obtaining sophisticated shapes. One of the existing solutions is electrical discharge machining using assisting techniques. Using assisting Cu-Ag and Cu mono- and multi-layer coatings of 40–120 μm and ZnO powder-mixed deionized water-based medium was proposed for the first time. The developed coatings were subjected to tempering and testing. It was noticed that Ag-adhesive reduced the performance when tempering had a slight effect. The unveiled relationship between the material removal rate, powder concentration, and pulse frequency showed that performance was significantly improved by adding assisting powder up to 0.0032–0.0053 $mm^3$/s for a concentration of 14 g/L and pulse frequency of 2–7 kHz. Further increase in concentration leads to the opposite trend. The most remarkable results corresponded to the pulse duration of 1 μs. The obtained data enlarged the knowledge of texturing insulating cutting ceramics using various powder-mixed deionized water-based mediums.

**Keywords:** alumina; assisting coating; brass wire; electrical discharge machining; sublimation; tempering

## 1. Introduction

The issues of micro-texturing of working surfaces of critical machine-building products made of difficult-to-machine materials, including those aluminum-based ceramics, have been the subject of special attention in recent years and the object of theoretical and experimental research by leading scientific groups [1–6]. Such cutting ceramics ($Al_2O_3$, AlN, SiAlON) exhibit exceptional thermomechanical and tribological properties making them unreplaceable materials for machining nickel-based alloys for aircraft and aerospace industries [7–11]. It should be noted that manufacturing cutting ceramic inserts involves processing a large number of microstructures and often requests additional coating [12–14]. Surface micromachining includes micro-profiling to create a three-dimensional specific relief on the surface, the dimensions and roughness of which are determined based on the operation characteristics and the textured material's physicochemical properties. These microtextures' functional purpose is to reduce the intensity of friction and adhesive setting between mating surfaces and increase the wear resistance and service life of the cutting tool many times under a wide range of operational loads [15–17]. The formed microrelief provides a significant reduction in the actual contact area. Additionally, it serves as micro-reservoirs for grease, microencapsulated lubricants, anti-friction materials, and liquids capable of forming and retaining anti-friction films between the contact surfaces for a long time [18,19].

Traditionally, diamond tools are used to process cutting ceramics. However, laser, micro-abrasive, water-jet machining, and chemical processing are well-known technologies of directed action on the surface layer [20–23]. Electrical discharge machining of dielectrics using assisting techniques is one of the promising ways of texturing the surface of products [24,25]. The advantages of this method include high reproducibility, accuracy, the ability to process complex geometry surfaces, high locality, and the ability to process superhard materials for which traditional mechanical methods are ineffective. The electrical discharge machining makes it possible to avoid time-consuming and expensive post-processing operations while maintaining the high quality of the workpiece. Due to the removal of material by electrical erosion that includes thermal and chemical dissociation of the material at a small distance between electrodes (the interelectrode gap is about 0.02 mm), the absence of physical contact between the tool and the workpiece is achieved. Consequently, the productivity of electrical discharge machining does not depend on the mechanical properties such as hardness, strength, and brittleness of the material to be processed but on its electrical properties. In addition, it is possible to increase the productivity of the process by reducing the processing time of complex geometry textures due to the possibility of using electrodes of various configurations or the technological mobility of the tool electrode with an accuracy of 80–100 nm.

It is well known that any material with a specific electrical conductivity above the threshold value of $10^5$ μΩ·cm can be subjected to electrical erosion. However, in 1986, Soviet scientists [26] invented and patented a method for electrical discharge machining of dielectrics. The method was further developed by leading scientists worldwide, developing its application concerning tool ceramics for 20 years [27–31].

The assisting electrode coating, which has proven itself in the works of many authors, is adhesive copper tape [32–35]. Insulating copper (II) oxide is unstable in the presence of hydrogen and reduces to conductive metallic copper [36–40]. At the same time, oxides of another copper-group metal, silver, such as $Ag_2O$ and $Ag_2O_2$, dissociate at temperatures above 280 °C and 100 °C, respectively, and exhibit conductive properties (specific electrical conductivity $\gamma$ of 60.0–62.5·$10^6$ S·m$^{-1}$) exceeding values for copper ($\gamma$ of 58.0–59.5·$10^6$ S·m$^{-1}$), making them even more attractive for developing technology of electrical discharge machining of insulating materials.

ZnO material was chosen for assisting powder due to the following reasons:

- It is commercially available;
- Stable under fire exposure conditions and is not reactive to water;
- Refers to the materials that require considerable preheating, under all ambient temperature conditions, before ignition and combustion can occur;
- Is a widely used *n*-type semiconductor;
- The band gap $E_g$ = 3.30–3.36 eV at room temperature [41–43]; the less the band gap, the more conductive properties the material exhibits [44,45]);
- Exhibits a chemical affinity for aluminum and copper [46–48].

It should be noted that ZnO powder is hazardous to aquatic organisms according to GHS Hazard pictograms; it is toxic and causes foundry fever if the dust is inhaled. It is coded by NFPA: Standard System for the Identification of the Hazards of Materials for Emergency Response as follows:

- Code 2: Intense or continued but not chronic exposure could cause temporary incapacitation or possible residual injury, for health;
- Code 1: Materials that require considerable preheating, under all ambient temperature conditions, before ignition and combustion can occur, for flammability;
- Code 0: Normally stable, even under fire exposure conditions, and is not reactive with water, for instability–reactivity.

However, it is relatively safe for the personnel working with micro-sized particles in suspension and actively used as additive of toothpaste and cement in therapeutic dentistry and cosmetic sunscreens.

Thus, developing electrical discharge machining techniques for increasing the productivity of electrical discharge machining of aluminum-based cutting ceramics in the case of study alumina using an assisting electrode coating and powder-mixed dielectric medium is relevant and in demand among modern tool production.

The object of research is electrical discharge machining of aluminum-based ceramics in the case of studying alumina using assisting techniques such as Cu-Ag and Cu mono- and multi-layer coating and ZnO powder-mixed deionized water medium. The subject of the study is the performance of electrical discharge machining of insulating alumina using a combined assisting electrode technique.

The study aims to find an alternative approach to electrical discharge machining of alumina and increase the productivity of insulating cutting ceramics machining using the assisting electrode technique by optimizing machining factors such as powder concentration, pulse frequency, and duration.

## 2. Materials and Methods

### 2.1. Sintering of the Samples

The corundum $\alpha$-$Al_2O_3$ A16SG (Alcoa, New York, NY, USA) was used for producing samples. The average particle diameter was 0.53 μm. A detailed description of powder characterization [49], preparing suspensions [50], drying, machining graphite molds of MPG-6 grade cold-pressed blanks using a carbide tool with a multi-layer combined PVD-coating [51–53] and diagnostic system [54–56], consequent pouring powder mixture and sintering [7,8,57–59] on a spark plasma sintering machine KCE FCT-H HP D-25 SD (FCT Systeme GmbH, Rauenstein, Germany) are presented in the previously published works. The sintered discs were 65.5 mm in diameter and 10 mm in thickness. Optical control was carried out on an Olympus BX51M instrument (Ryf AG, Grenchen, Switzerland). Scanning electron microscopy of the obtained kerfs and their chemical analyses were conducted on a VEGA3 instrument (Tescan, Brno, Czech Republic). For each kerf, at least five spectra scannings were produced.

### 2.2. Electrical Discharge Machining

A two-axis wire electrical discharge machine ARTA 123 Pro (NPK "Delta-Test", Fryazino, Russia) was used for experiments (Table 1). The open tank system of the machine allows using any deionized water- or oil-based dielectric medium out of the filtration system [60].

**Table 1.** Main characteristics of wire electrical discharge machine ARTA 123 Pro.

| Parameters | Value and Description |
|---|---|
| Max axis motions $X \times Y \times Z$, mm | $125 \times 200 \times 80$ |
| Tool positioning accuracy, μm | $\pm 1$ |
| Average surface roughness parameter $R_a$, μm | 0.6 |
| Dielectric medium | Any |
| Max power consumption, kW | <6 |

A brass wire-electrode of 0.25 mm in diameter made of CuZn35 brass provided texture formation on alumina without taking into account the spark gap (path offset). The preliminary testing in a deionized water medium reduced the range of the machining factors. The range of the chosen factors is presented in Table 2. The choice of the factors is based on [60]. However, the range of factors was intentionally extended for exploratory research. The previously conducted work with alumina showed that the optimal value of the operational current is 0.3–0.4 A [49]. The main correlation between current, operational voltage, and character of the obtained wells on the surface or material removal rate is as follows [61]:

$$\sum F_{imp} = I \cdot U_o, \tag{1}$$

where $\Sigma F_{imp}$ is the summarized force of working impulses in the system's action, N; $I$ is current, A; $U_o$ is operational voltage, V.

**Table 2.** Range of electrical discharge machining factors.

| Factor | Measuring Units | Value |
|---|---|---|
| Operational voltage, $U_o$ | V | 108; 72; 60; 48; 36 |
| Pulse frequency, $f$ | kHz | 2; 5; 8; 11; 15; 17; 20; 25; 30 |
| Pulse duration, $D$ | μs | 0.5; 1; 1.5; 1.75; 2; 2.35; 2.5; 2.68; 2.7 |
| Rewinding speed, $v_W$ | m/min | 3; 3.4; 7; 10 |
| Feed rate, $v_F$ | mm/min | 0.1; 0.3; 0.4; 0.5; 1 |
| Wire tension $F_T$ | N | 0.05; 0.1; 0.25; 0.3; 0.4 |

Machining is carried out according to the control program of the translational movement of the wire-electrode along the *X*-axis from the zero position to a depth of 0.98 mm (taking into account the spark gap) [62–64]. The development of factors was carried out following the results of pre-implemented experiments for each type of the developed assisting coating. Each failed experiment minimized the range of the factors and assisted in determining the optimal values in terms of the material removal rate (productivity) [65].

Adaptive control based on voltage control in the gap (control of factors) is necessary to avoid contact between the tool electrode and the conductive assisting coating that leads to a short circuit. When electric shortcuts are registered in the gap, the wire tool is retracted automatically to establish the required interelectrode gap and ensure effective machining when the number of working pulses (aimed at destroying the workpiece or assisting coating) exceeds the number of idle pulses (aimed at destroying erosion products, debris). The effective ratio of working pulses to the number of impulses is 0.7–0.9 [66–68]. It should be noted that voltage in the gap and concentration of conductive debris strongly influence surface roughness (density of formed wells) when the current influences their overall size [69–73]. Further improvement in roughness is not observed when the concentration exceeds a certain level. Moreover, with increased concentration, the more frequent appearance of shorts can be observed until the wire tool is blocked in the kerf, and further machining is not possible. At the same time, the concentration of non-conductive particles reduces the process productivity and leads as well to the blockage of the wire tool.

The coated blank was fixed on the machine table during experiments (Figure 1). The basing was carried out by wire tool approach along the *X*- and *Y*- axes; the surface of the coated workpiece was taken as zero. The wire tool was adjusted vertically along the *Z*-axis by a spark. The position +2–+3 mm from the assisting coating was taken as zero of the wire tool. The upper nozzle was placed at +2–+3 of the coated workpiece to ensure adequate flushing by turbulent flows [74,75]. Electrical discharge machining was carried out with immersion of the workpiece in the ZnO-powder-mixed deionized water medium. The dielectric fluid level was established 1–2 mm above the workpiece. The workpiece was held for 8–10 min in a dielectric before machining to avoid the influence of thermal fluctuations. The obtained sample was wiped with a rag over [76]. At least 5 kerfs were produced for each parameters' set.

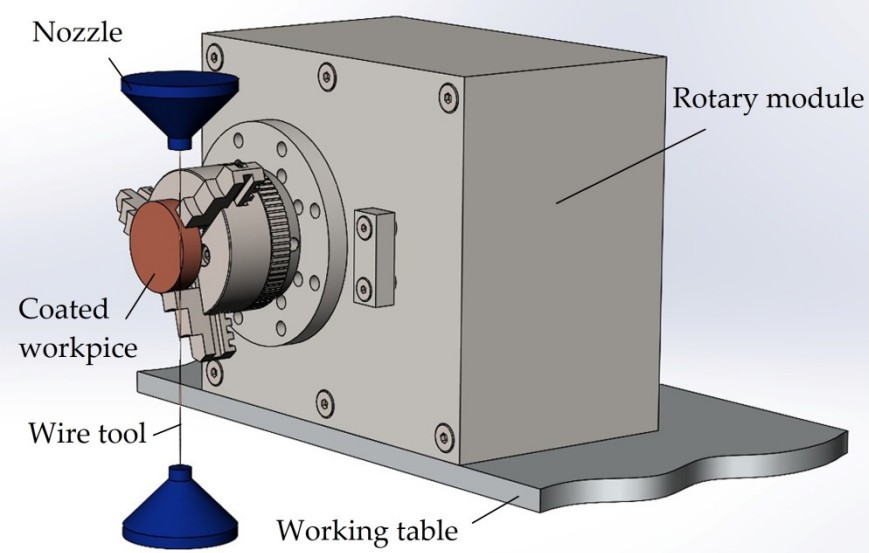

**Figure 1.** Scheme of coated workpiece microtexturing by wire electrical discharge machining.

Formed textures (kerfs) were controlled optically. Here and below, the optical measurement error was calculated by the formula [77–81]:

$$\delta_l = \pm 3 + \frac{L}{30} + \frac{g \cdot L}{4000}, \tag{2}$$

$$\delta_t = \pm 3 + \frac{L}{50} + \frac{g \cdot L}{2500}, \tag{3}$$

where $\delta_l$ is the longitudinal measurement error, μm; $\delta_t$ is the transversal measurement error, μm; $L$ is the measured length, mm; $g$ is the product height above microscope table glass (taken equal to zero), mm.

The material removal rate (*MRR*) was calculated as follows [82]:

$$MRR = \frac{V}{t}, \tag{4}$$

where the volume of removed material is calculated using the formula:

$$V = S \cdot l, \tag{5}$$

where $S$ is the kerf area in the plan, mm$^2$; $l$ is the length of the kerf, μm. A detailed description of calculating $S$ is provided in [50]. The processing time $t$ is calculated from the feed rate $v_F$ of the electrode-tool and workpiece height (kerf length) $h$:

$$t(s) = \frac{h(mm)}{v_F\left(\frac{mm}{s}\right)} \tag{6}$$

### 2.3. Assisting ZnO-Powder-Mixed Deionized Water Medium

Deionized water (LLC "Atlant", pos. Marusino, Lyubertsy district, Moscow region, The Russian Federation) following ASTM D-5127-90 with specific electrical resistivity up to 18.0 MΩ·cm was chosen as a suspension basis to avoid the formation of insulating $Al_4C_3$ or $Al_2(C_2)_3$ [83,84].

ZnO-powder-mixed deionized water medium was tested at concentrations of 7, 14, 21, 35, 50, and 100 g/L to improve the performance of the developed system. A total of $76 \times 5$ experiments were carried out. The zinc oxide ZnO of "Ch" grade, 99% of purity (LLC "Unihim", Saint Petersburg, The Russian Federation), following GOST 10262-73, bulk

density of 5.61 g/cm$^3$, was used for producing suspension. The chemical composition is presented in Table 3.

**Table 3.** Chemical composition of ZnO powder ("Ch" grade, 99% of purity).

| Chemical Substances | Chemical Formula | wt.% |
|---|---|---|
| Zinc oxide | ZnO | Balance |
| Manganese | Mn | ≤0.0005 |
| Arsenic | As | ≤0.0002 |
| Cadmium | Cd | not standardized |
| Potassium permanganate | KMnO$_4$ | ≤0.01 |
| Potassium | K | ≤0.005 |
| Calcium | Ca | ≤0.01 |
| Substances insoluble in hydrochloric acid | - | ≤0.01 |
| Sulfates | SO$_4$ | ≤0.01 |
| Phosphates | PO$_4$RR′R″ | not standardized |
| Chlorides | Cl$_x$R (x = 1–5) | ≤0.004 |
| Iron | Fe | ≤0.001 |
| Sodium | Na | not standardized |
| Copper | Cu | ≤0.001 |
| Lead | Pb | ≤0.01 |

ZnO is a colorless crystalline powder, insoluble in water, turning yellow when heated, and subliming at 1800 °C [85–88]. Zinc oxide is a direct-gap semiconductor with a band gap $E_g$ of 3.30–3.36 eV [41–43]. Natural doping with oxygen makes it an *n*-type semiconductor. When heated, the substance changes color: white at room temperature, and zinc oxide becomes yellow. This is explained by a decrease in the band gap and a shift of the edge in the absorption spectrum from the UV region to blue.

The powder was subjected to granulometric analysis and optical microscopy. An EL104 laboratory balance (Mettler Toledo, Columbus, OH, USA) with a measurement range of 0.0001–120 g weighed powder with an error of 0.0001 g. ZnO powder was sifted using an analytical sieving machine AS200 basic (Retsch, Dusseldorf, Germany) with a test sieve (10 μm by ISO 3310-1). The previously conducted studies showed that the smallest size of suspended particles led to the highest productivity [69–73].

The prepared suspension was constantly stirred during experiments. For the higher powder concentration, electrical discharge machining was intensified by ultrasonic vibrations using an ultrasonic unit IL100-6/1 (LLC "Ultrasonic Technology—INLAB", Saint Petersburg, Russia) to avoid powder conglomerations in the discharge gap at the higher particle concentration (100 g/L) at frequency 22 kHz [89]. It should be noted that higher than 30 kHz and up to 1 MHz could be harmful to the biological process in the human body since arising cavitation with bubble formation with a diameter of less than 1 μm (ultrasound surgery). The emitting tip was placed at the tank to provide ultrasonic vibrations in volume. As known, the speed of propagation of vibrations in an elastic body is much higher than in a liquid or gaseous medium. Thus, the suspension is subjected to bulk ultrasonic vibrations.

After processing, the samples were cleaned with alkali.

*2.4. Assisting Electrode*

A few types of copper-based assisting coatings were developed (Table 4, Figure 2a). A HomaFix 404 20 m/10 mm copper tape (JSC Electroma, Lipetsk, The Russian Federation) was used as a basis for the developed coatings (Figure 2b). The main properties of the tape are provided in Table 5. A conductive polymer-based silver-containing adhesive (synthetic resins) Kontaktol (Keller, Yekaterinburg, The Russian Federation), with an electrical resistance of 10$^{-6}$ Ω·m (γ of 10$^6$ S·m$^{-1}$) was used in the sandwich-type of assisting coating electrode forming a complete continuous uniform coating of variable thickness of 20–200 μm on the sample. The ratio of silver powder to acetone is (120–140)/(40–60). The

recommended operating temperature is up to +160 °C. The adhesive coating was deposed using a brush. Half of the samples were additionally kept in an oven (drying cabinet) at temperatures of +160, +200, and +240 °C for 60, 90, 120, and 180 min to ensure the drying of the polymer base of the tape, removing organic soluble media and reducing stress.

**Table 4.** Developed assisting coatings.

| Assisting Coating | Adhesive Type | Thickness, mm | Electrical Conductivity $\gamma$ [1], $S \cdot cm^{-1}$ | Specific Electrical Resistivity [2], $\Omega \cdot mm^2 \cdot m^{-1}$ |
|---|---|---|---|---|
| Silver Adhesive | Polymer-based + Silver powder | 0.100–0.110 | $0.009486 \pm 0.00001$ | $1.0542 \times 10^{-6}$ |
| Copper tape, 1 layer | Resin-based | 0.040 | | |
| Copper tape, 2 layers | Resin-based | $2 \times 0.040$ | $0.580046 \pm 0.00001$ | $0.01724 \times 10^{-6}$ |
| Copper tape, 3 layers | Resin-based | $3 \times 0.040$ | | |
| Sandwich "Copper tape + Silver Adhesive", 1 layer | Polymer-based + Silver powder | 0.150 | | |
| Sandwich "Copper tape + Silver Adhesive", 2 layers | Polymer-based + Silver powder | $2 \times 0.150$ | $0.584112 \pm 0.00001$ | $0.01712 \times 10^{-6}$ |
| Sandwich "Copper tape + Silver Adhesive", 3 layers | Polymer-based + Silver powder | $3 \times 0.150$ | | |
| Graphite [3] | - | - | - | 8.00 |
| Distilled water [3] | - | - | - | $10^3–10^4$ |

[1] experimental values; [2] calculated values; [3] for reference.

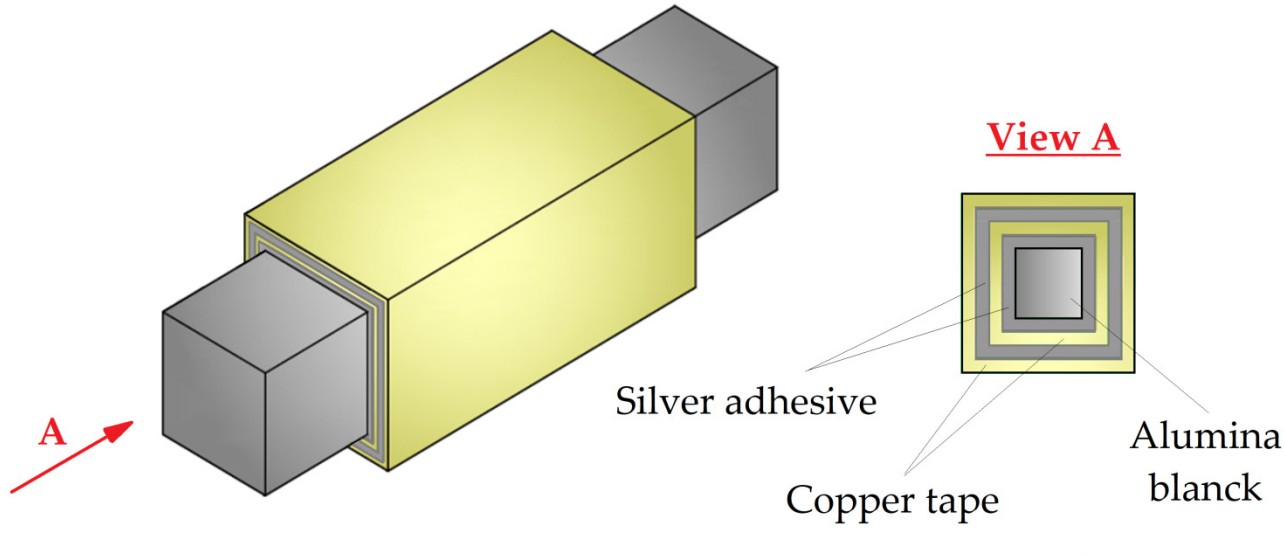

(a)

**Figure 2.** *Cont.*

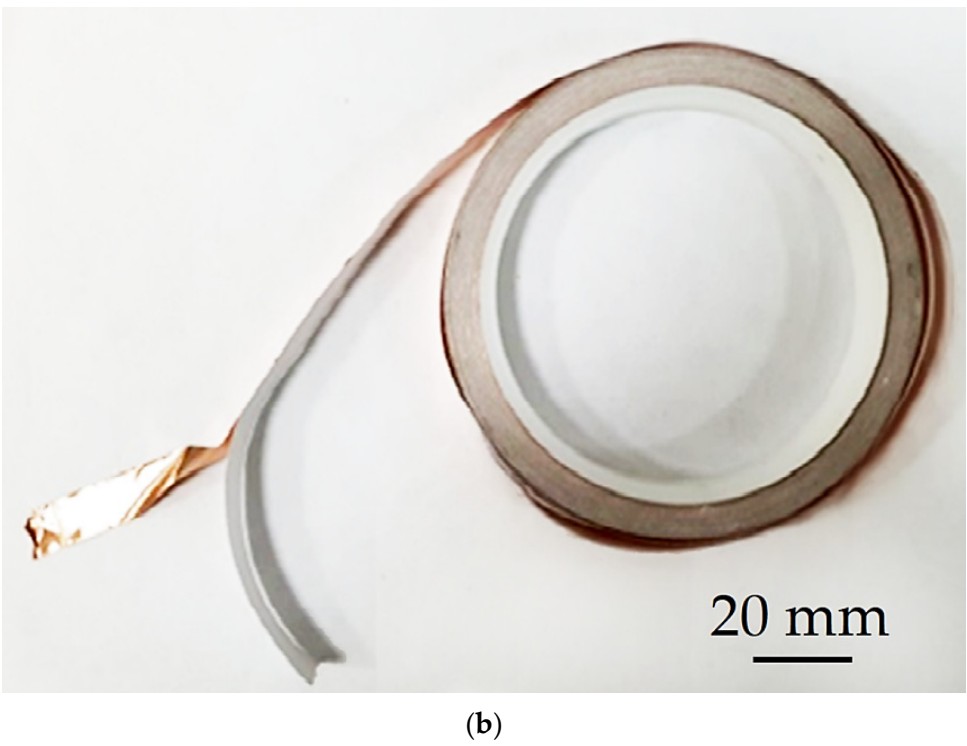

(**b**)

**Figure 2.** (**a**) Developed sandwich-type coating (schematic presentation of Sandwich "Copper tape + Silver Adhesive", 2 layers); (**b**) Copper tape.

**Table 5.** Parameters of copper tape.

| Parameter | Value |
|---|---|
| Thickness of copper basis, mm | $0.035 \pm 0.0002$ |
| Tensile strength, N/cm | 115 |
| Elongation (Extension ratio), % | <2 |
| Specific electrical resistivity, $\Omega \cdot mm^2 \cdot m^{-1}$ | 0.016–0.017 |
| Operating temperature, °C | From $-40$ to $+110 \pm 5$ |
| Tape width, mm | 10 |

The specific electrical resistance of the formed assisting coatings (Table 3) was controlled by a Fischer Sigmascope SMP10 device (Helmut Fischer GmbH, Sindelfingen, Germany). The thickness of the first layer of the developed coatings was controlled with a Calowear instrument (CSM Instruments, Needham, MA, USA). It was developed to carry out wear tests on a small scale using the spherical notch method. In other words, it forms a recess by erasing the sample material during the rotation of a ball of a certain diameter (20 mm) coated with an abrasive medium (spherical microabrasion method). The method has proven to be fast for analyzing the thickness of any coating (mono- or multi-layer) and determining the wear coefficient of massive materials and coatings [90–92].

Before coating, the ceramic samples were placed in an ultrasonic tank and cleaned using a soap solution at a temperature of 60 °C for 20 min and alcohol for 5 min [93–95]. Approbation of the coated samples was conducted in a deionized water medium. Coating removal was conducted by a complex method: washing in an ultrasonic tank and mechanical cleaning.

## 3. Results

### 3.1. Characterization of ZnO Powder

Granulomorphometric analysis and optical microscopy (Figure 3) of the ZnO powder showed that the powder sample had an average inner diameter of 63.79 µm and 33.46 µm

for 50% of the particles, while an average area diameter of 80.06 μm and 47.19 for 50% of the particles. The zinc oxide particles have a larger diameter with a high percentage of particles. The average circularity of zinc oxide powder is about 0.597 μm and about 0.625 μm for 50% of the particles. The powder was sieved before further processing.

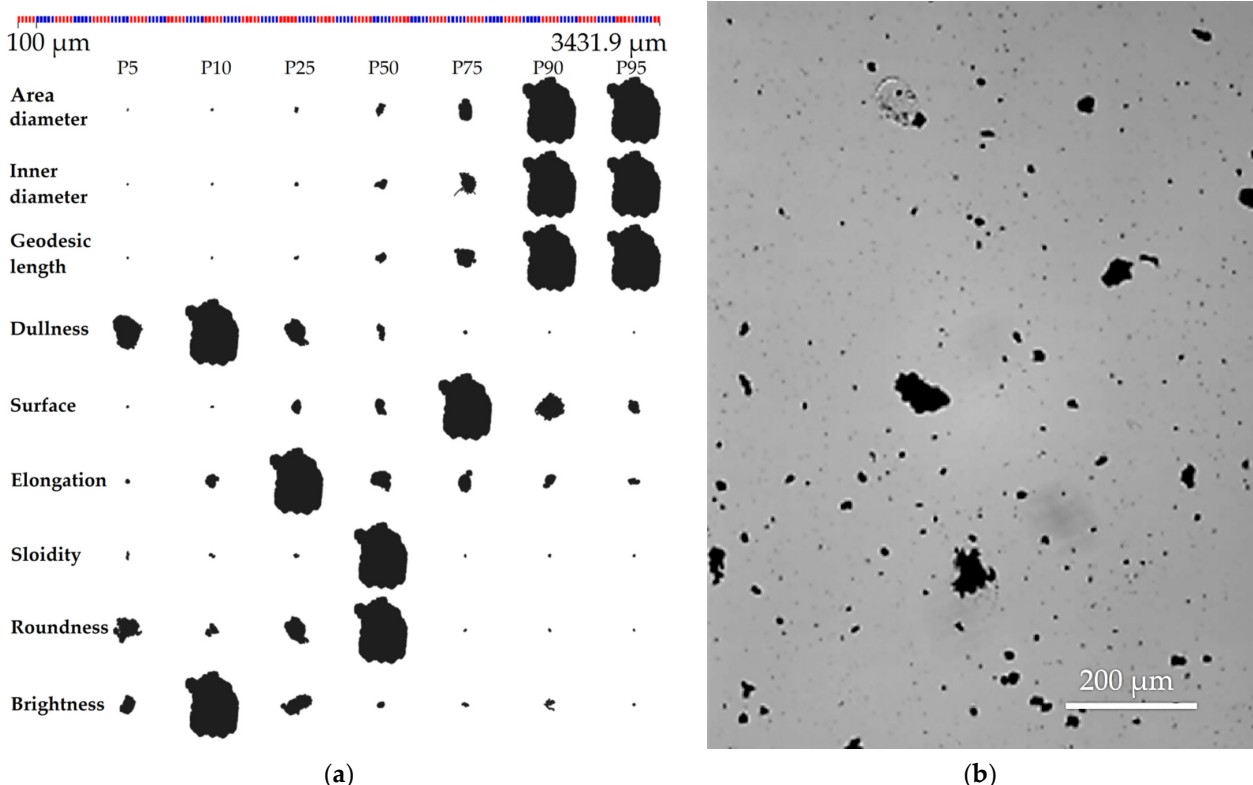

(**a**)            (**b**)

**Figure 3.** Granulomorphometric analysis of the composition of ZnO powder (**a**) and morphology (**b**).

### 3.2. Preliminary Testing of the Developed Coatings in Deionized Water Medium

Figure 4a shows the optical microscopy of kerf after electrical discharge machining the samples coated with silver adhesive. The traces of the coating erosion are observed. However, the traces of ceramic workpiece sublimation were not remarked (absence of the erosion of ceramics under discharge pulses). In this context, the term "sublimation" is used for the electrical erosion of ceramics since:

The alumina's boiling point is about 2980 °C [96], and the temperature in the discharge spark is about 10,000 °C [97–99]. With such a difference between the boiling point and surrounding temperature in a pulse period of 1–100 μs under conditions of continuously during the pulse expanding rarefied low-temperature gas-plasma bubble (region of low pressure) (Figure 4b), the material cannot pass the stages from solid to liquid, vapor, and plasma steadily, and direct sublimation of the material occurs from solid to vapor and ion plasma state [100].

It should be noted that a detailed phase diagram of the state of aluminum oxide at elevated and reduced pressure requires additional research and is still not presented in the literature, as well as for many other substances. An analysis of the preliminary testing results for sandwich types assisting coatings showed that the holding time in the oven does not significantly affect the adhesion of the copper tape to ceramics and material removal rate for the whole range of the developed coatings (Figure 4c). Differences in adhesion between two and three-layer coatings (80 and 120 μm in thickness) were also not observed. Approbation of samples of the "silver + copper" sandwich coatings showed similar results with variations with low tempering modes and multi-layer structure. The difference (Figures 5 and 6) does not have a fundamental nature to the purposes of the current study.

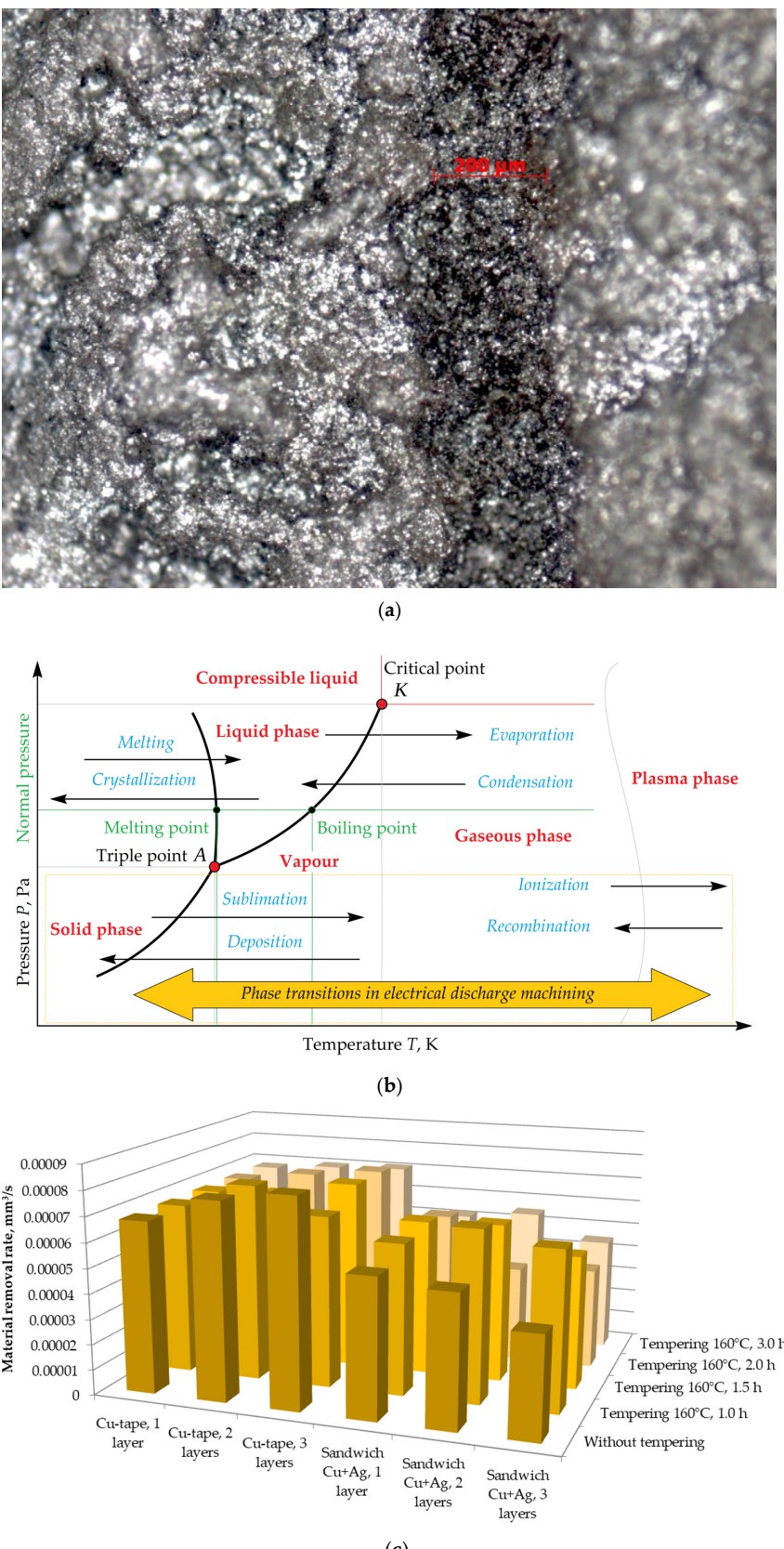

**Figure 4.** (**a**) The preliminary testing results of the electrical discharge machining of samples coated with a silver adhesive (optical microscopy), 10×; (**b**) phase transitions and states of oxide ceramics; (**c**) the preliminary testing results of the electrical discharge machining of alumina using Cu-Ag and Cu mono- and multi-layer coatings and various coating tempering modes.

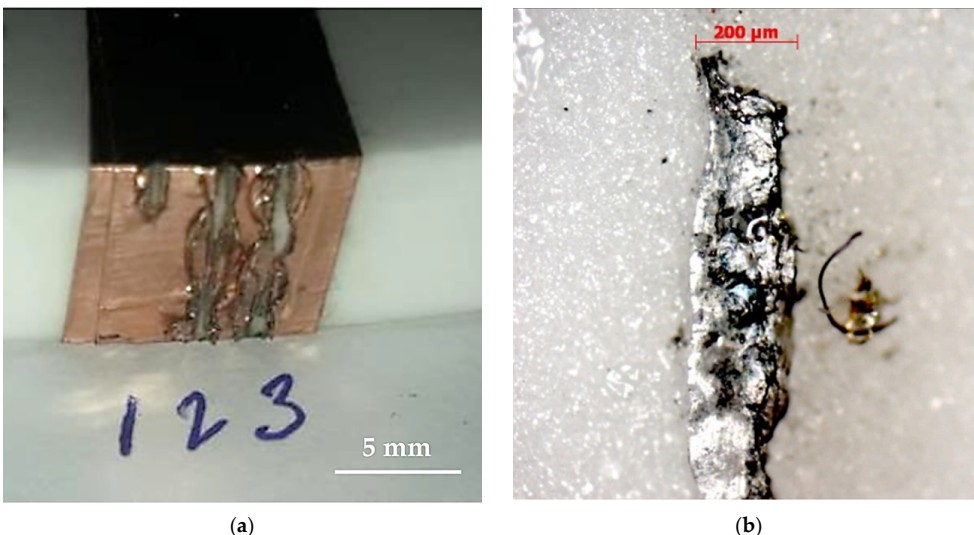

(**a**)                                                                                     (**b**)

**Figure 5.** The preliminary testing results of the electrical discharge machining of samples coated with self-adhesive copper tape: (**a**) a sample after preliminary testing; (**b**) optical microscopy of the formed texture, 10×.

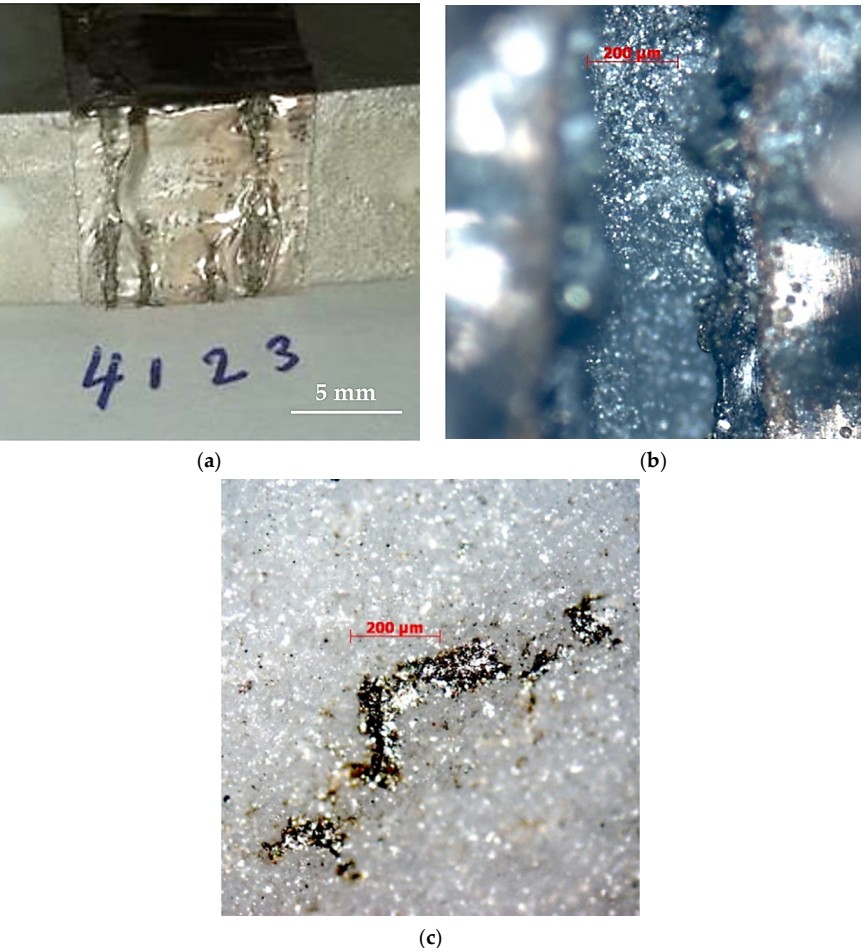

(**a**)                                                                                     (**b**)

(**c**)

**Figure 6.** The preliminary testing results of the electrical discharge machining of samples with a sandwich-type mono-layer coating "silver + copper" after an hour of tempering at a temperature of 160 °C: (**a**) a sample after preliminary testing; (**b**) optical microscopy of the formed texture before coating removal, 10×; (**c**) optical microscopy of the formed texture, 10×.

The samples after the preliminary testing were cleaned. The deposition of metallic copper and its adhesion to the ceramic base is observed after cleaning. The use of a sandwich-type coating visually improved the adhesion of the coating (Figure 6) but significantly reduced the effect of the coating when electrical discharge machining compared with the sample with self-adhesive copper tape (Figure 5). The reduction in effect was consistent with the expected decline in the bulk conductivity of the coating due to the use of synthetic resin-based adhesives. Further experiments were conducted with the double-layer copper tape coating without silver adhesive (copper tape, 2 layers). The range of the machining factors was reduced (Table 6).

**Table 6.** Range of electrical discharge machining factors.

| Factor | Measuring Units | Value |
| --- | --- | --- |
| Operational voltage, $U_o$ | V | 108 |
| Pulse frequency, $f$ | kHz | 2; 5; 7; 10; 15; 20; 25; 30 |
| Pulse duration, $D$ | µs | 0.5; 1.0; 1.5; 2.0; 2.5; 2.64 |
| Rewinding speed, $v_W$ | m/min | 7 |
| Feed rate, $v_F$ | mm/min | 0.3 |
| Wire tension $F_T$ | N | 0.25 |

*3.3. Electrical Discharge Machining in ZnO Powder-Mixed Deionized Water Medium*

The general view of the kerfs and results of experiments obtained by optical microscopy for the various powder concentration, pulse frequencies, and duration are shown in Figure 7. During conducting experiments with the powder-mixed deionized water medium at a concentration of 7 g/L, the wire tool was interrupted at a frequency of 30 kHz, pulse duration from 1.0 to 2.64 µs, except for the experiment with a pulse duration of 2.5 µs (Figure 7a). There were little erosion marks at a frequency of 5, 10, 15, and 30 kHz and a pulse duration of 2.5 µs. The non-stable erosion results were remarked for pulse frequency of 5–25 kHz and pulse duration of 1.0, 1.5, 2.0 µs and pulse frequency of 7 and 25 kHz and pulse duration of 2.5 µs.

At a concentration of 14 g/L, the effect of powder adding was visually reduced: the most pronounced result was achieved at a frequency of 7 kHz and a pulse duration of 1.0 µs (Figure 7b). Non-stable erosion marks were noticed for a pulse frequency of 5, 10 kHz and a pulse duration of 1.0, 1.5 µs. Little erosion marks were observed at a pulse frequency of 7, 10 kHz and a pulse duration of 1.5, 2.0, 2.5 µs.

At a concentration of 21 g/L, the effect of powder addition was also visually reduced. The most pronounced result was achieved at a pulse frequency of 5 and 7 kHz and a pulse duration of 0.5 and 1.0 µs, respectively (Figure 7c). Non-stable erosion marks were noticed for:

- A pulse frequency of 7 and 10 kHz and a pulse duration of 0.5 µs;
- A pulse frequency of 5 and 10 kHz and a pulse duration of 1.0 µs.

Little erosion marks were observed at a pulse frequency of 5 and 10 kHz and a pulse duration of 1.5 µs. It was noticed that visually electrical discharge machining is more stable at a pulse duration of 1.0 µs.

At a concentration of 35 g/L, the most pronounced result was achieved at a pulse frequency of 5 and 10 kHz and a pulse duration of 1.0 µs (Figure 7d). Little erosion marks were observed at a pulse frequency of 7 kHz and a pulse duration of 1.0 µs.

At a concentration of 50 g/L, the most pronounced result was achieved at a pulse frequency of 2 and 10 kHz and a pulse duration of 1.0 µs (Figure 7e). Little erosion marks were observed at a pulse frequency of 5, 7, and 15 kHz and a pulse duration of 1.0 µs.

At a concentration of 100 g/L, the most pronounced result was achieved at a pulse frequency of 2, 7, and 10 kHz and a pulse duration of 1.0 µs (Figure 7f). Non-stable erosion marks were observed at a pulse frequency of 5 kHz and a pulse duration of 1.0 µs.

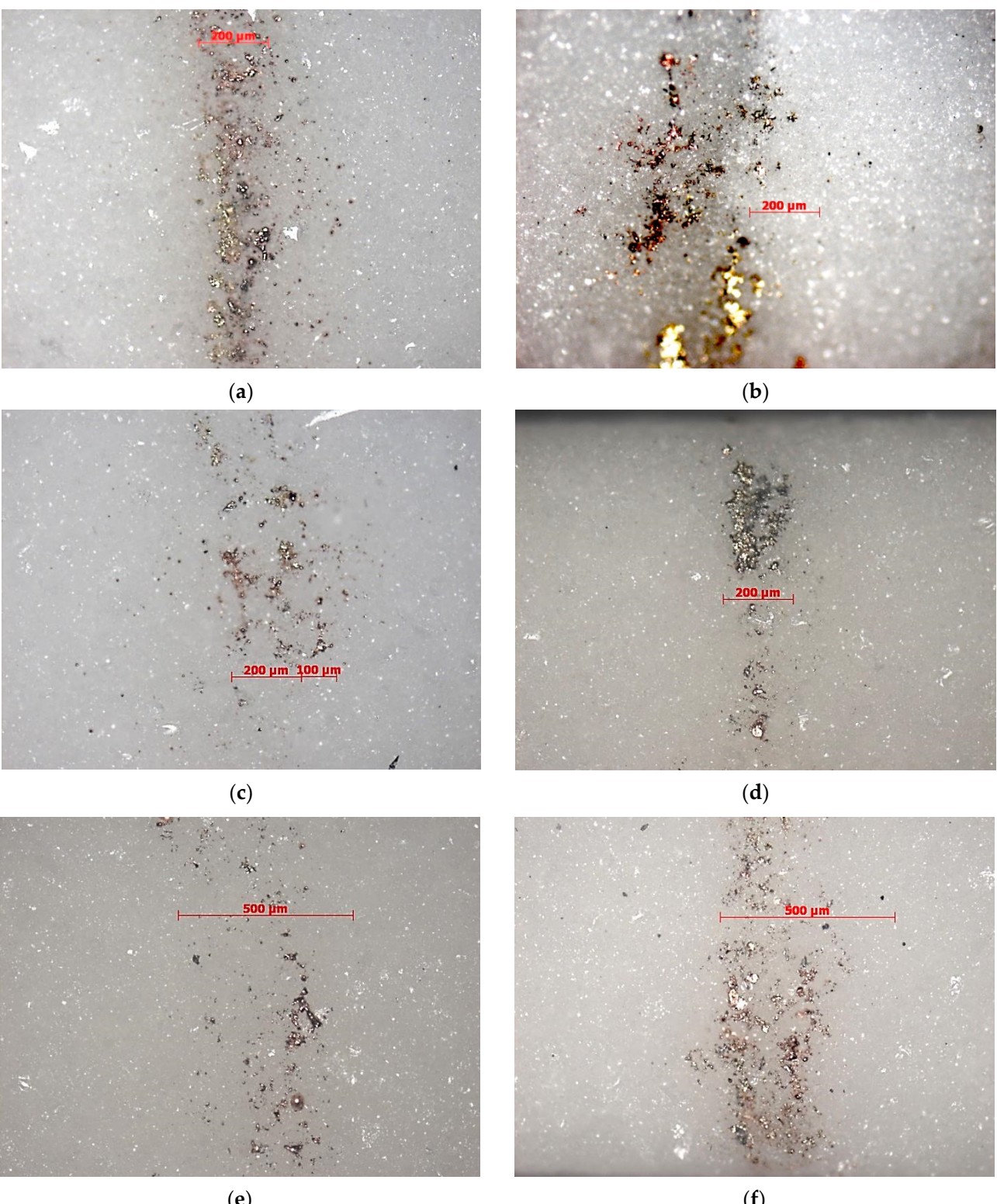

**Figure 7.** Optical microscopy of the obtained kerf in $Al_2O_3$ samples coated with a double-layer copper tape coating of 80 µm (general view): (**a**) concentration of 7 g/L, pulse frequency $f$ = 5 kHz, pulse duration $D$ = 1.0 µs; (**b**) concentration of 14 g/L, pulse frequency $f$ = 5 kHz, pulse duration $D$ = 1.0 µs; (**c**) concentration of 21 g/L, pulse frequency $f$ = 5 kHz, pulse duration $D$ = 0.5 µs; (**d**) concentration of 35 g/L, pulse frequency $f$ = 5 kHz, pulse duration $D$ = 1.0 µs; (**e**) concentration of 50 g/L, pulse frequency $f$ = 5 kHz, pulse duration $D$ = 1.0 µs; (**f**) concentration of 100 g/L, pulse frequency $f$ = 7 kHz, pulse duration $D$ = 1.0 µs.

As can be seen, the effect of adding powder is noticeable at powder concentrations of 7 and 14 g/L (Figure 7a,b). The effect is visually reduced at a powder concentration of 21–50 g/L (Figure 7c–e) and appears again at a powder concentration of 100 g/L (Figure 7f). The technological gaps are presented in Table 7.

**Table 7.** Technological gaps of electrical discharge machining alumina in ZnO powder-mixed deionized water medium.

| Powder Concentration, g/L | Pulse Frequency, kHz | Pulse Duration, µs | | | | | |
|---|---|---|---|---|---|---|---|
| | | 0.5 | 1.0 | 1.5 | 2.0 | 2.5 | 2.64 |
| 7 | 2 | x | x | x | x | x | x |
| | 5 | x | 1 | 1 | 1 | 0 | x |
| | 7 | x | 1 | 1 | 1 | 1 | x |
| | 10 | x | 1 | 1 | 1 | 0 | x |
| | 15 | x | 1 | 1 | 1 | 0 | x |
| | 20 | x | 1 | 1 | 1 | x | x |
| | 25 | x | 1 | 1 | 1 | 1 | x |
| | 30 | x | x | x | x | 0 | x |
| 14 | 2 | x | x | x | x | x | x |
| | 5 | x | 1 | 1 | x | x | x |
| | 7 | x | 2 | 1 | 0 | 0 | x |
| | 10 | x | 1 | 0 | 0 | 0 | x |
| | 15 | x | x | x | x | x | x |
| | 20 | x | x | x | x | x | x |
| | 25 | x | x | x | x | x | x |
| | 30 | x | x | x | x | x | x |
| 21 | 2 | x | x | x | x | x | x |
| | 5 | 2 | 1 | 0 | x | x | x |
| | 7 | 1 | 2 | x | x | x | x |
| | 10 | 1 | 1 | 0 | x | x | x |
| | 15 | x | x | x | x | x | x |
| | 20 | x | x | x | x | x | x |
| | 25 | x | x | x | x | x | x |
| | 30 | x | x | x | x | x | x |
| 35 | 2 | x | 0 | x | x | x | x |
| | 5 | 0 | 2 | 0 | x | x | x |
| | 7 | x | 1 | x | x | x | x |
| | 10 | 0 | 2 | 0 | x | x | x |
| | 15 | x | x | x | x | x | x |
| | 20 | x | x | x | x | x | x |
| | 25 | x | x | x | x | x | x |
| | 30 | x | x | x | x | x | x |
| 50 | 2 | 0 | 2 | 0 | x | x | x |
| | 5 | x | 1 | x | x | x | x |
| | 7 | x | 1 | x | x | x | x |
| | 10 | 0 | 2 | 0 | x | x | x |
| | 15 | x | 1 | x | x | x | x |
| | 20 | x | x | x | x | x | x |
| | 25 | x | x | x | x | x | x |
| | 30 | x | x | x | x | x | x |
| 100 | 2 | x | 2 | 0 | x | x | x |
| | 5 | x | 1 | 0 | x | x | x |
| | 7 | 0 | 2 | 0 | x | x | x |
| | 10 | 0 | 2 | x | x | x | x |
| | 15 | x | 0 | x | x | x | x |
| | 20 | x | x | x | x | x | x |
| | 25 | x | x | x | x | x | x |
| | 30 | x | x | x | x | x | x |

NB: x is the absence of the traces, 0 is little erosion marks, 1 is non-stable erosion marks, 2 is the most pronounced result.

### 3.4. Scanning Electron Microscopy and Chemical Analyses

The results of scanning electron microscopy (SEM) in secondary electrons and qualitative and quantitative analysis of the obtained kerfs after electrical discharge machining of alumina using a copper assisting coating are shown in Figure 8. SEM-image (Figure 8a) shows the workpiece's microstructure with the deposed material and the traces of the material's sublimation (thermal dissociation). Figure 8b demonstrates a uniform distribution of aluminum and oxygen and localization of the deposed copper in the kerf. The energy-dispersive spectroscopy of the deposed material (Figure 8c) demonstrates the prevalence of copper in the spectra. The carbon corresponds to the normal atmospheric contamination of the samples. The presence of the formed oxides is partly related to the normal contamination of the samples and partly related to the material of the insulating ceramic (bound with aluminum). The presence of aluminum is minor. A quantitative analysis of the five spectra is presented in Table 8.

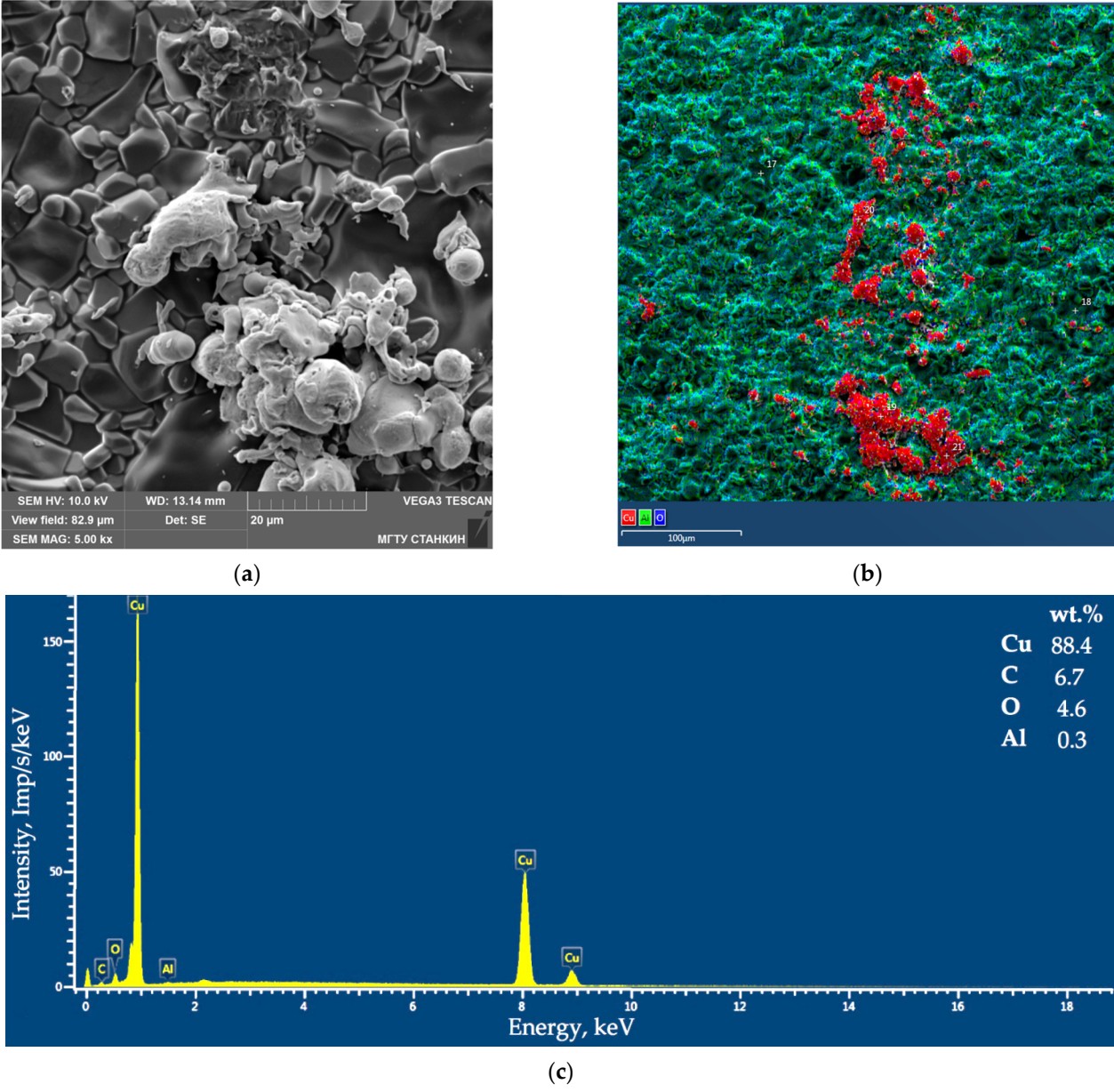

**Figure 8.** SEM-analyses of the machined kerf in the $Al_2O_3$ sample: (**a**) image in secondary electrons, 5.00 k×; (**b**) chemical mapping, 1.00 k×; (**c**) energy dispersive spectroscopy of the deposed material.

**Table 8.** Chemical analysis of the kerf.

| Spectrum | Weight Ratio of O, wt.% | Weight Ratio of Al, wt.% | Weight Ratio of Cu, wt.% | Weight Ratio of C, wt.% |
|---|---|---|---|---|
| 1 | 54.19 | 45.81 | - | - |
| 2 | 54.95 | 45.05 | - | - |
| 3 | 2.43 | - | 86.68 | 10.88 |
| 4 | 3.41 | 1.48 | 84.08 | 11.03 |
| 5 | 4.64 | 0.3 | 88.37 | 6.69 |

## 4. Discussion

A graphical presentation of the calculated material removal rate, which shows the relationship between material removal rate, pulse frequency, and concentration of ZnO-suspension for a pulse duration of 1 µs, is shown in Figure 9. The optimized values of the factors of electrical discharge machining alumina using a double-layer copper tape assisting coating of 80 µm in thickness with ZnO powder-mixed water medium of 14 g/L were: pulse frequency of 5 and 7 kHz, pulse duration of 1 µs.

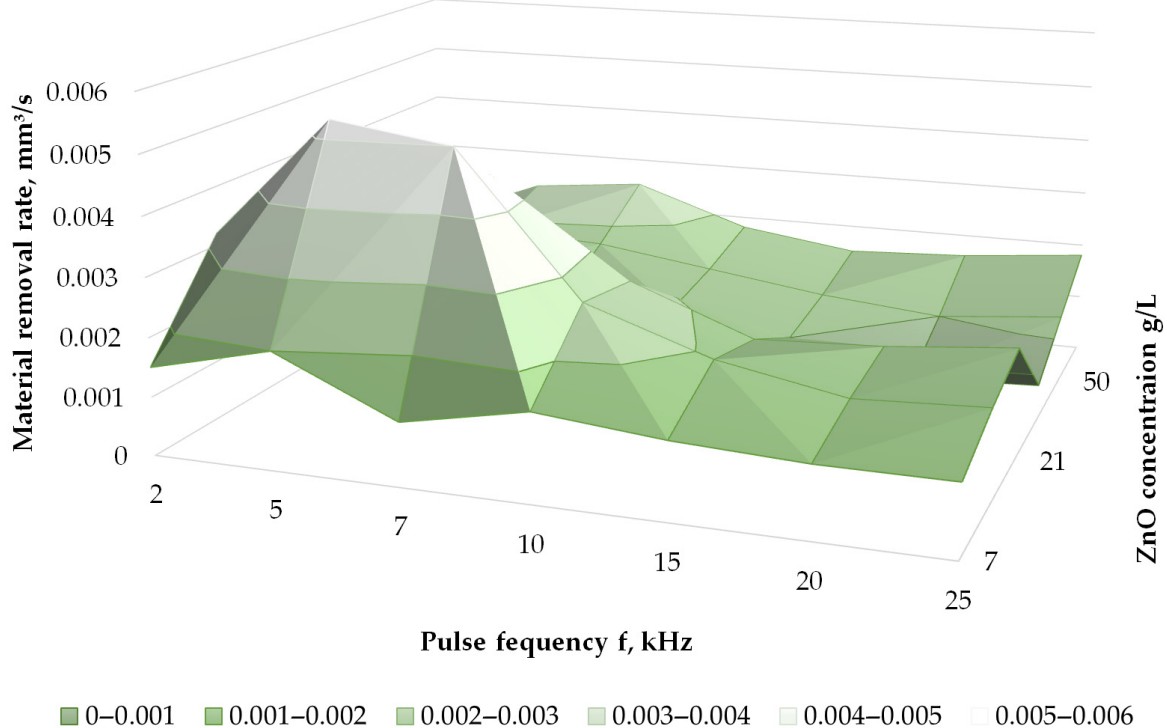

**Figure 9.** The electrical discharge machining alumina performance using a double-layer copper tape assisting coating of 80 µm in thickness with ZnO powder-mixed deionized water-based dielectric medium for a pulse duration of 1 µs.

In comparison with the previously published data, the achieved performance does not extend analogs [33,49,50,69], where the maximum achievable performance was 0.0084 $mm^3/s$ [50] for deionized water-based medium and 0.0213 $mm^3/s$ for hydrocarbons [69] but significantly enlarged the current knowledge on electrical discharge machining of insulating ceramics as follows:

- Even with the lower concentration of the suspension powder, it is possible to achieve higher values of material removal rate (7 g/L for ZnO in the current study comparing 150 g/L for $TiO_2$ [50]) in combination with a copper coating that can be related to the electrical properties of the assisting powder such as band gap more than to specific electrical resistance of the coating;

- Complex multi-layer complex coating can reduce performance due to a decrease in the electrical conductivity: there is a noticeable difference between mono- and multi-layer Cu-Ag and Cu sandwich coatings and a mono- and double-layer coatings, but no effect was observed between double and triple copper coating;
- Temperature and holding time during coating tempering do not demonstrate any noticeable effect. However, a slight improvement was observed between tempered and not tempered samples that do not have a principle character but significantly enlarge the labor intensity of the work.

## 5. Conclusions

The paper solves a scientific and technical problem of electrical discharge machining of insulating alumina using assisting mono- and multi-layer Cu-Ag and Cu coatings of 40–120 μm and ZnO powder-mixed deionized water-based medium. Using mono- and multi-layer Cu-Ag and Cu coatings and ZnO powder-mixed water medium was proposed for the first time.

The research showed that tempering temperature and holding time have an insignificant effect on the electrical discharge machining of alumina performance, while multi-layer coating visually improves the performance compared to mono-layer coating. At the same time, there is no visible difference between double- and triple-layer coatings and using silver adhesive reduces the effect of the assisting coating.

The conducted work established a relationship between the material removal rate, powder concentration, and pulse frequency. Using ZnO-powder significantly improves performance up to noticeable values of 0.0015–0.0020 $mm^3$/s for a concentration of 7 g/L and pulse frequency of 2–5 kHz and 0.0032–0.0053 $mm^3$/s for a concentration of 14 g/L and pulse frequency of 2–7 kHz. Further concentration increase leads to the opposite trend: at a concentration of 100 g/L, a slight increase in performance is observed (0.0023–0.0025 g/L) for a pulse frequency of 5–7 kHz. The most remarkable results corresponded to the pulse duration of 1 μs.

The obtained data enlarge knowledge on texturing insulating cutting ceramics using various types of powder-mixed deionized water-based mediums and can be used for producing a new class of cutting inserts for machining nickel-based alloys.

**Author Contributions:** Conceptualization, M.A.V.; methodology, A.A.O.; software, K.I.G.; validation, K.H.; formal analysis, K.H.; investigation, A.A.O.; resources, K.H.; data curation, K.I.G.; writing—original draft preparation, A.A.O.; writing—review and editing, A.A.O.; visualization, K.H. and K.I.G.; supervision, M.A.V.; project administration, M.A.V.; funding acquisition, M.A.V. All authors have read and agreed to the published version of the manuscript.

**Funding:** This work was supported financially by the Ministry of Science and Higher Education of the Russian Federation (project No FSFS-2021-0006).

**Institutional Review Board Statement:** Not applicable.

**Informed Consent Statement:** Not applicable.

**Data Availability Statement:** Data are available in a publicly accessible repository.

**Acknowledgments:** The study was carried out on the equipment of the Center of collective use of MSUT "STANKIN" supported by the Ministry of Higher Education of the Russian Federation (project No. 075-15-2021-695 from 26.07.2021, unique identifier RF-2296.61321X0013).

**Conflicts of Interest:** The authors declare no conflict of interest.

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
