# Peer review of "Electrical Discharge Machining of Alumina Using Cu-Ag and Cu Mono- and Multi-Layer Coatings and ZnO Powder-Mixed Water Medium"

_technologies, doi:10.3390/technologies11010006_

Round 1

Reviewer 1 Report

The article "Electrical Discharge Machining of Alumina using Cu-Ag and Cu Multi-Layer Coating and ZnO Powder-Mixed Water Medium" reported a new method to EDM alumina with Cu-Ag and Cu coatings and ZnO powder mixing water based medium. The article is well organised and included all important elements. 

To further improve the manuscript, please consider the following:

1) Typos were detected. example line 74 -temperates, table 7, weight ratio (r missing)

2)  define h of eq 8 (line 177).

3) line 258, please further explain how to reach this 'there are no traces of ceramic workpiece sublimation.' (line 258).

4) "table 6" and "4 discussion", 7 pulse durations were studied but it is unsure why 1 mirco second is the optimum.

5) Fig 9,  Please explain how to read "the relationship between material removal rate, pulse frequency and duration, and concentration of ZnO-suspension, " from Fig 9?

6) It is unsure how to obtain "Temperature and holding time during coating tempering do not demonstrate any noticeable effect. However, a slight improvement was observed between tempered and not tempered samples that do not have a principle character but significantly enlarge the labor intensity of the work." from your experimental work. 

Author Response

Response to Reviewer 1 Comments

Dear reviewer,
Thank you so much for your kind evaluation of our work. We do agree with all your proposals and comments and have modified the manuscript according to them.

We hope that the manuscript will be suitable for publishing in Technologies and will attract many potential readers of the journal with your comments. The introduced corrections in the text of the manuscript are marked yellow.

Kind regards,
Authors.

Reviewer comments
Point 1: Typos were detected. example line 74 -temperates, table 7, weight ratio (r missing)

Response 1: Thank you for pointing it out. It is revised, and the text was checked once again.

Point 2: define h of eq 8 (line 177).

Response 2: Thank you for noticing it. The explanation is added.

Point 3: line 258, please further explain how to reach this 'there are no traces of ceramic workpiece sublimation.' (line 258).

Response 3: Thank you, we agree that it was no clear remarked. The additional explanation is added:

However, the traces of ceramic workpiece sublimation were not remarked (absence of the erosion of ceramics under discharge pulses). In this context, the term “sublimation” is used for the electrical erosion of ceramics since:

  • The alumina’s boiling point is about 2 980°C, and the temperature in the discharge spark is about 10 000°C. With such a difference between the boiling point and surrounding temperature in a pulse period of 1-100 µs under conditions continuously during the pulse of an expanding gas-plasma discharged bubble (region of low pressure) (Figure 4b), the material cannot pass the stages steadily from solid to liquid, vapor, and plasma, and, consequently, direct sublimation of the material occurs from solid to vapor and plasma state. It should be noted that a detailed phase diagram of the state of aluminum oxide at elevated and reduced pressure requires additional research and is still not presented in the literature, as well as for many other substances.

Point 4: "table 6" and "4 discussion", 7 pulse durations were studied but it is unsure why 1 mirco second is the optimum.

Response 4: Thank you for this remark. With other pulse frequency values, non-stable electrical erosion marks were observed. An additional explanation is added to the text.

Point 5: Fig 9,  Please explain how to read "the relationship between material removal rate, pulse frequency and duration, and concentration of ZnO-suspension, " from Fig 9?

Response 5: Thank you for pointing it out. It is revised. In the current version, it is for a pulse frequency of 1 µs.

Point 6: It is unsure how to obtain "Temperature and holding time during coating tempering do not demonstrate any noticeable effect. However, a slight improvement was observed between tempered and not tempered samples that do not have a principle character but significantly enlarge the labor intensity of the work." from your experimental work.

Response 6: Thank you; it was mentioned in subsection 3.2. An additional explanation is added.

Reviewer 2 Report

1.     Include latest paper published in year 2022 on the studied topic.

2.     Remove all typo and grammatical mistakes: Line 14 airspace-aerospace, replace water as coolant with deionized water, remove discontinuity from the first paragraph of the introduction and this issue is there in the whole manuscript, line 61-62 sentence is not clear, line 163;

3.     English proofreading is required.

4.     How ZnO is safe, the claim is not right.

5.     Line 127 need reference.

6.     Why the tested parameters in table 2 are selected for the current work?

7.     Does the authors take the ratio of ZnO based on certain standards, needs an explanation.

Author Response

Response to Reviewer 2 Comments

Dear reviewer,

Thank you so much for your kind evaluation of our work. We agree with all your proposals and comments and have modified the manuscript accordingly.

We hope the manuscript will be suitable for publishing in Technologies and attract many potential journal readers with your comments. The introduced corrections in the text of the manuscript are marked green.

Kind regards,

Authors.

Reviewer comments
Point 1: Include latest paper published in year 2022 on the studied topic.

Response 1: Thank you so much for your kind recommendation. We have considered a few recent papers on powder-mixed electrical discharge machining:

No.

Title

Authors

Year

Journal

1

Multi-objective Optimization of EDM and Powder Mixed EDM for H-11 Steel

Tripathy, S., Tripathy, D.K.

2023

Lecture Notes in Mechanical Engineering

pp. 689-698

2

Multi-objective optimization of powder-mixed EDM parameters using hybrid Grey-ANFIS artificial intelligence technique

Singh, J.

2022

International Journal on Interactive Design and Manufacturing

16(4), pp. 1533-1549

3

Machinability of Nimonic Alloy 90 in µ-Titanium Carbide Mixed Electrical Discharge Machining

Sundaresan, D., Marappan, L., Thangavelu, K., Venkatraman, V.

2022

Arabian Journal for Science and Engineering

47(12), pp. 15223-15243

4

Machining performance and sustainability analysis of PMEDM process using green dielectric fluid

Bajaj, R., Tiwari, A.K., Pramanik, A., Srivastava, A.K., Dixit, A.R.

2022

Journal of the Brazilian Society of Mechanical Sciences and Engineering

44(11),563

5

Analysis on the Performance of Micro and Nano Molybdenum Di-Sulphide Powder Suspended Dielectric in the Electrical Discharge Machining Process—A Comparison

Rajesh, J.V., Abimannan, G.

2022

Nanomaterials

12(20),3587

6

Experimental Investigations and Effect of Nano-Powder-Mixed EDM Variables on Performance Measures of Nitinol SMA

Chaudhari, R., Shah, Y., Khanna, S., (...), Pimenov, D.Y., Giasin, K.

2022

Materials

15(20),7392

Most of them have no relation to the discussed in the paper topic since using ANOVA, hydrocarbon fluid, and non-ceramic workpieces. We have considered one of the articles that we found attractive, but due to unknown or published unproven information, it cannot be quoted (5). Firstly, the dielectric type is not mentioned in the paper when the choice of powder concentration of 1 g/L was not founded. The conclusion includes the following:

At larger magnitudes of discharge duration (>300 μs), owing to the lessened energy intensity inside the compressed vapour bubble, the melted material is not completely expelled from the spark vicinity, which leads to a reduced crater depth.

However, that was never proven in the article and seemed to be the result of the authors’ fantasy since there is no “melted material” in the “vapor bubble” at a temperature of 10 000°C achieved in the discharge channel but the plasma state of the material direct sublimated from the workpiece surface that afterward deposed in the form of drops on the electrode surfaces. The article includes as well a false equation (5): R(t)=2040 × Ip 0.43 × Td 0.44, which can be easily checked since micrometers are not the result of multiplying amperes (peak current) by microseconds (discharge duration) and even in strange and non-obvious powers. The real equation would have a form of Euler’s number raised to powers close to the following, like most processes, the relationship of input factors and output parameters of technology:

where β and t are factor and coefficient with inverse dimensions, and R0 is the initial value in micrometers as R(t) [https://doi.org/10.3390/met10111540].

Other disadvantages of this article are that there needs to be more discussion of the safety of the MoS2 powder and the pulse frequency. Meanwhile, MoS2 dissociates with oxygen and hydrogen up to MoO3 + SO2 and Mo + H2S, respectively, when heated higher than 400C and 800C. MoO3 dissociates as well when heated to Mo + H2O. H2O is active to MoS2 at 500C, and forms MoO2 + H2S. Resulting SO2 and H2S are toxic. SO2 dissolves in water to form sulfurous acid, H2S is explosive. All these should have been taken into account when writing and reviewing manuscripts.

In addition use of vibration with a frequency higher than 30 kHz and up to 1 MHz could be harmful to the biological process in the human body since arising cavitation with bubble formation with a diameter of less than 1 µm (ultrasound surgery) [Andreeva, T.A.; Berkovich, A.E.; Bykov, N.Y.; Kozyrev, S.V.; Lukin, A.Ya. High-Intensity Focused Ultrasound: Heating and Destruction of Biological Tissue. Technical Physics 2020, 65, 1455-1466; https://link.springer.com/article/10.1134/S1063784220090030 ]; the works should be conducted according to the sanitary norms and rules of production.

Point 2: Remove all typo and grammatical mistakes: Line 14 airspace-aerospace, replace water as coolant with deionized water, remove discontinuity from the first paragraph of the introduction and this issue is there in the whole manuscript, line 61-62 sentence is not clear, line 163;

Response 2: Thank you for pointing it out:

  • “Aerospace” is revised, as it is indeed more related to the industry when airspace is a designated part of the sky for aircraft use;
  • We have replaced “water” in the text of the manuscript with “deionized water” (mostly in the context with medium);
  • “discontinuity” needs more explanation, we are sorry that we did not get the point. Is it more grammatical “discontinuity” or linguistic one? We have tried to make the paragraphs bigger where it seems too short and not logical. Unfortunately, it is difficult to see with the correction mode in Word, but we hope it can be seen when the corrections will be accepted. Or is it related to the sign “hyphen” to break (wrap) and hyphenate words to another line? We have turned it off but we are afraid that it was as it is in layout (we hope that the editorial office will help to organize it correctly according to the requirements of Technologies);
  • Lines 61-62 are rewritten;
  • Line 163, the sentence is revised (standard instructions for EDM for after-processing maintenance).

Point 3: English proofreading is required.

Response 3: Thank you for your kind suggestion. The text was revised by a native speaker. Additionally, it was subjected to a grammar check. The report is attached. We hope that the current version looks better to attract potential readers of the journal.

Point 4: How ZnO is safe, the claim is not right.

Response 4: Thank you for pointing it out. It probably was unclear from the context: relatively safe when working with micro-sized particles in suspension. To avoid misunderstanding, the relevant passage is added. We would add that it is also used as a dietary supplement in breakfast cereals (a source of zinc).

Point 5: Line 127 need reference.

Response 5: Thank you for noticing it; the relevant references have been added.

Point 6: Why the tested parameters in table 2 are selected for the current work?

Response 6: Thank you for a good question. Actually, the choice of the factors is based on previously conducted work [Grigoriev, S.N.; Volosova, M.A.; Okunkova, A.A.; Fedorov, S.V.; Hamdy, K.; Podrabinnik, P.A.; Pivkin, P.M.; Kozochkin, M.P.; Porvatov, A.N. Electrical Discharge Machining of Oxide Nanocomposite: Nanomodification of Surface and Subsurface Layers. J. Manuf. Mater. Process. 2020, 4, 96]. However, the range of factors was intentionally extended for exploratory research. The relevant sentences are added.

Point 7: Does the authors take the ratio of ZnO based on certain standards, needs an explanation.

Response 7: Thank you for noticing it. As a basis for the exploratory research, we have taken a concentration of 7 - 10 g/L from [66]. Since we used another powder (no graphite for the same material as the workpiece), it was decided to go from 7 g/L to a certain point when suspension still can be called liquid medium. At 100-150 g/L, the desired effect of suspension was achieved. Further increase in concentration (200 g/L) significantly hampered the EDM machine's maintenance and the tool electrode's free move. The relevant passage is added.

Round 2

Reviewer 2 Report

Dear Authors, Thanks a lot for incorporating the suggestions and explaining it a very precise and organized way.

English proofreading is required